# Markers of Bronchiolitis Obliterans Syndrome after Lung Transplant: Between Old Knowledge and Future Perspective

**DOI:** 10.3390/biomedicines10123277

**Published:** 2022-12-17

**Authors:** Dalila Cavallaro, Marco Guerrieri, Stefano Cattelan, Gaia Fabbri, Sara Croce, Martina Armati, David Bennett, Antonella Fossi, Luca Voltolini, Luca Luzzi, Alberto Salvicchi, Piero Paladini, Adriano Peris, Miriana d’Alessandro, Paolo Cameli, Elena Bargagli, Tuscany Transplant Group, Laura Bergantini

**Affiliations:** 1Respiratory Disease and Lung Transplant Unit, Department of Medical Sciences, Surgery and Neurosciences, University of Siena, 53100 Siena, Italy; 2Thoracic Surgery Unit, Department of Experimental and Clinical Medicine, Azienda Ospedaliero-Universitaria Careggi, 50134 Firenze, Italy; 3Lung Transplant Unit, Department of Medical, Surgical and Neuro Sciences, Azienda Ospedaliero-Universitaria Senese, University of Siena, 53100 Siena, Italy; 4Organizzazione Toscana Trapianti, Regione Toscana, 50134 Firenze, Italy

**Keywords:** biomarkers, lung transplant, bronchiolitis obliterans syndrome

## Abstract

Bronchiolitis obliterans syndrome (BOS) is the most common form of CLAD and is characterized by airflow limitation and an obstructive spirometric pattern without high-resolution computed tomography (HRCT) evidence of parenchymal opacities. Computed tomography and microCT analysis show abundant small airway obstruction, starting from the fifth generation of airway branching and affecting up to 40–70% of airways. The pathogenesis of BOS remains unclear. It is a multifactorial syndrome that leads to pathological tissue changes and clinical manifestations. Because BOS is associated with the worst long-term survival in LTx patients, many studies are focused on the early identification of BOS. Markers may be useful for diagnosis and for understanding the molecular and immunological mechanisms involved in the onset of BOS. Diagnostic and predictive markers of BOS have also been investigated in various biological materials, such as blood, BAL, lung tissue and extracellular vesicles. The aim of this review was to evaluate the scientific literature on markers of BOS after lung transplant. We performed a systematic review to find all available data on potential prognostic and diagnostic markers of BOS.

## 1. Introduction

A lung transplant is considered the best therapeutic option for patients with end-stage lung disease. Survival after lung transplant continues to improve: median survival is now 6.9 years thanks to the efficacy of prophylaxis, new drugs, and better risk stratification [1]. However, graft failure is responsible for 22.7% of deaths in the interval between 30 days and 1 year after transplant. After the first year, chronic lung allograft dysfunction (CLAD) is the leading cause of death [2]. The international society for heart and lung transplantation (ISHLT) consensus recently re-defined CLAD and related phenotypes [3,4]. CLAD is defined as a persistent decline (≥20%) in the measured forced expiratory volume in the first second (FEV1) from the post-transplant baseline. The date of CLAD onset is defined as the date of the first FEV1 decline ≤80% of baseline [4].

Bronchiolitis obliterans syndrome (BOS), restrictive allograft dysfunction (RAS) and the newly defined mixed-phenotype result are the three major phenotypes of CLAD. BOS is the most common form of CLAD and is characterized by airflow limitation and an obstructive spirometric pattern unexplained by acute rejection, infection or other coexistent condition without high-resolution computed tomography (HRCT) evidence of parenchymal opacities [4,5,6]. 

Obliterative bronchiolitis (OB) occurs in a number of clinical settings, including infection, inhalational injury, drug toxicity or in the context of a collagen-vascular disease. There are also idiopathic cases [7]. OB is a clinical syndrome marked by progressive dyspnea and cough with the absence of parenchymal lung disease. Among lung transplant recipients, “bronchiolitis obliterans syndrome,” is described as a disorder with clinical and histopathological similarities to OB, representing the leading cause of long-term allograft dysfunction and mortality [8].

Although less frequent, RAS and mixed phenotypes portend a worse prognosis than BOS [9].

Computed tomography and micro-CT analysis show abundant small airway obstruction, starting from the fifth generation of airway branching and affecting up to 40–70% of airways [10]. The pathogenesis of BOS remains unclear. It is a multifactorial syndrome that leads to pathological tissue changes and clinical manifestations [11]. BOS is mediated by several mechanisms, such as alloimmune reactivity, humoral and autoimmunity, activation of innate immune cells and response to nonimmune-related allograft insults, such as infection and aspiration [12]. Employing a systematic approach for the collection of clinical data and the storage of samples can be useful for improving the knowledge of BOS research [13,14]. 

Because BOS is associated with the worst long-term survival in LTx patients, many studies are focused on the early identification of BOS [15]. Markers may be useful for diagnosis [16,17,18] and for understanding the molecular and immunological mechanisms involved in the onset of BOS [19,20]. Diagnostic and predictive markers of BOS have also been investigated in various biological materials, such as blood, BAL, lung tissue and extracellular vesicles [21,22,23,24].

The aim of this review was to evaluate the scientific literature on markers of BOS after lung transplant. We performed a systematic review to find all available data on potential prognostic and diagnostic markers of BOS. We performed a systematic search of PUBMED scientific papers in the English language. We used the keywords “markers of bronchiolitis obliterans syndrome” and “markers of CLAD” to locate potentially relevant studies. Reference lists of the studies selected were also checked to obtain additional sources. We excluded reviews, case reports, unrelated topics and studies in which BOS was not diagnosed according to the most recent guidelines. One of the 86 articles screened was excluded as it was a duplicate. Of the remaining 85, 14 were excluded because they did not meet the inclusion criteria, 11 were reviews and 3 were case reports. Of the remaining articles, 40 were excluded as unrelated topics, 1 because it concerned the paediatric population, 5 because they were based on guidelines before 2015 and 1 due to an inadequate (very small) cohort. In the end, 31 articles were selected for our study. A flowchart of selected articles is shown in Figure 1.

### Molecular and Immunological Mechanisms Involved in the Pathogenesis of BOS

The molecular and immunological mechanisms underlying BOS are unclear, and no disease-specific biomarkers have yet been found [25]. From the histopathological point of view, BOS is characterized by the accumulation of a submucosal extracellular matrix, muscle cell hyperplasia and complete obliteration of the airway lumen with partial destruction of the original smooth muscle layer. Chronic inflammation occurs in these patients and leads to excessive recruitment and/or activation of (myo-)fibroblasts in small peripheral airways [26].

Evidence of aberrant angiogenesis has also been reported in bronchiolitis obliterans lesions, consisting of the proliferation and enlargement of the microvasculature [14,15]. This mechanism could explain the flow limitation and dyspnoea typical of patients with BOS [27,28,29]. 

Antibodies also play a crucial role in the development of BOS. Graft antigen antibodies are closely linked to the development of BOS in lung transplant patients because graft-reactive antibodies induce activation of the complement system and degradation of lung tissue [30].

Concerning immunity, an increase in Th1 cells or cytokine-related Th1 in blood or BAL fluid of BOS patients suggests that Th1 cells play a role in the process of CLAD [16,17]. Th17 seems to be involved in the development of BOS, although the immunological mechanisms are largely unexplored. Th17 supports chronic inflammation that may favour chronic dysfunction through airway fibrosis, neutrophil chemotaxis and/or expansion of autoantibodies [31]. Moreover, an imbalance in Th17 and regulatory T cells, resulting in an increased Th17/Treg ratio, is linked to chronic dysfunction. Inflammatory factors in a BOS microenvironment can favour the differentiation of Treg into Th17 cells through the production of IL-6, modifying the Th17/Treg ratio and facilitating chronic dysfunction [32]. 

Concerning the role of B cells, no clear molecular mechanisms leading to chronic rejection have been established. An accumulation of B cells is observed in the lung tissue of patients with CLAD [33], and the presence of donor-specific HLA antibodies is related to BOS development [21,22,34,35,36,37].

Toll-like receptors (TLR) have a central role in the pathogenesis of BOS. Infections, ischemia time and ischemia-reperfusion injury can be activated by these receptors. Colonisation by pathogens such as *Aspergillus fumigatus* and *Pseudomonas aeruginosa* can also stimulate TLR and chemokine production (CXCL1 and CXCL5) [23,24]. The selected articles are reported in Table 1. 

## 2. Tissue Markers

The study of tissue from lung transplant patients can be useful for investigating the expression of genes, proteins and molecules specific to BOS lesions. Although biopsy is an invasive procedure, tissue specimens can provide information about molecular and immunological aspects.

### 2.1. Liver Kinase B1 Gene

Liver kinase B1 (*LKB1*) is a serine/threonine protein kinase 11 implicated in tumour suppression and in the regulation of cell metabolism [69]. A major function of *LKB1* is the activation of 5′ AMP-activated protein kinase (AMPK), a regulator of metabolism and cell growth, which plays an inhibitory role in the epithelial-mesenchymal transition, tissue fibrosis and malignant transformation [70]. *LKB1* was originally identified as the product of a loss of function mutation in different genetic syndromes and malignancies [71]. In non-small cell lung cancer, loss of *LKB1* is associated with a more aggressive phenotype of this tumour [72,73]. Low expression of *LKB1* is associated with a higher risk of organ rejection and with the occurrence and progression of human chronic graft-versus-host disease after bone marrow transplant [74]. A recent study [38] compared *LKB1* activities in biopsies from newly diagnosed BOS and stable lung transplant patients, demonstrating significant downregulation of the *LKB1* gene in BOS [38]. In line with the literature, the results of the study suggest that the downregulation of *LKB1* may lead to tissue fibrosis, facilitating the development of BOS and that this protein could, therefore, be a reliable biomarker of risk of BOS.

### 2.2. Checkpoint Molecules

Different cell subsets, including dendritic cells, macrophages and T cells, as well as anti-HLA antibodies and interleukins, are implicated in chronic rejection and are involved in the expression and regulation of immune checkpoints [75,76,77]. The role of immune checkpoints in the development of BOS has been partially investigated. 

Programmed death 1 (PD1) and cytotoxic T-lymphocyte-associated protein 4 (CTLA-4) are proteins expressed on the surface of T and B cells that contribute to the down-regulation of the immune system while promoting self-tolerance by suppressing T cell inflammatory activity [78,79]. The ligand of PD-1, named PDL1, binds PD1, suppressing the effector function of T cells [80]. The role of immune checkpoints has been studied widely as a targeted therapy in cancer [81].

The possible role of the expression and function of immune checkpoint molecules in chronic allograft dysfunction is not clear. Sporadic experiences with immune checkpoint inhibitor treatments of kidney and heart transplant patients with cancer have shown that administration of these molecules results in the rapid development of severe rejection [82]. Righi et al. used immunohistochemistry [46] to detect significantly lower PD-1-, PDL1- and CTLA4 in BOS patients than in those with RAS. The exhausted phenotype of PD-1 cells, expressed in T cells and regulatory T cells (Tregs), proved to be significantly lower in RAS than in BOS patients. The triggers that chronically stimulate Tregs can determine their loss of function. This may contribute to the uncontrolled immune response that characterizes BOS.

### 2.3. VEGF/VEGFR2

Vascular endothelial growth factor (VEGF) is a strong angiogenic factor essential for angiogenesis. VEGF is active in physiological functions such as bone formation, haematopoiesis and wound healing [83]. It is produced by different cell lines, including macrophages, keratinocytes and fibroblasts. Although VEGF is essential for physiological homeostasis in various cell lines and tissues, it is also important in the pathogenesis of tumour growth and metastasis since it mediates vascular permeability and neo-angiogenesis [84,85,86,87]. VEGF and VEGF receptor 2 (VEGFR2) have been implicated in pulmonary vascular remodelling and in chronic rejection [88,89].

In this regard, a recent study [63] analysed concentrations of VEGF produced by distal-derived lung fibroblasts, before and after stimulation with TGF-β, in LTx patients with BOS and in stable LTx recipients and healthy controls, at baseline and follow-up (3, 6 and 12 months). It emerged that VEGF synthesis from distal-derived lung fibroblasts was significantly lower 3 months after lung transplant than in non-transplanted subjects and was increased at 6 months and 12 months after transplant, achieving the same level as in the healthy controls. These findings demonstrate the processes of the remodelling of pulmonary and bronchial vessels after lung transplant. Stimulation with TGF-β significantly enhanced the synthesis of VEGF in fibroblasts from lung transplant patients and healthy non-transplanted subjects. After TGF-β1 stimulation, levels of VEGF in fibroblasts obtained from patients who developed BOS 12 months after LTx were significantly lower than in patients without chronic rejection. The number of cells expressing markers of collagen production, VEGFR2/p4OH, was higher in patients with BOS, suggesting that BOS is related to chronic inflammation and subsequent fibrosis.

### 2.4. CXCL9 and CXCL10

C-X-C motif chemokine ligands 9 and 10 (*CXCL9* and *CXCL10*) are chemokines stimulated by IFN-gamma that induce chemotaxis and promote differentiation and multiplication of leukocytes. They have also been associated with a wide spectrum of inflammatory lung diseases through pro- and antifibrotic effects [90]. Common mechanisms of rejection have been reported in different grafted organs, leading researchers to combine data from multiple organs and formulate a common rejection module consisting of 11 genes (*CD6*, *TAP1*, *CXCL10*, *CXCL9*, *INPP5D*, *ISG20*, *LCK*, *NKG7*, *PSMB9*, *RUNX3* and *BASP1*) overexpressed during allograft rejection [91]. In recently published data, chronic rejection has been investigated by comparing the expression of these genes in lung tissue of BOS and RAS patients [66], showing lower expression of *TAP1*, *CXCL9* and *CXCL10* in BOS than in RAS patients. The results of the study suggest that the expression of these genes could be a useful marker to identify BOS.

## 3. Exhaled Breath

Exhaled breath is a useful, easy to obtain and non-invasive method in clinical practice. It is a source of information about mechanisms occurring in the alveoli and small airways. Little is yet known about the role of exhaled biomarkers in the development and pathogenesis of BOS.

### 3.1. Nitric Oxide

Nitric oxide (NO) is an important mediator involved in chronic inflammation of the lung [92]. At the lung level, NO acts as a vasodilator, bronchodilator, neurotransmitter and inflammatory mediator. High levels of exhaled nitric oxide have been reported in LTX patients with acute rejection, infection and lymphocytic bronchiolitis.

The NO produced by the respiratory system includes alveolar concentrations of NO (CalvNO) and maximum conducting airway wall flux (J’awNO), an expression of bronchial NO [93]. Concentrations of NO in exhaled breath can be investigated by non-invasive measurement of FeNO (fraction of exhaled nitric oxide). FeNO is measured during slow exhalation from total lung capacity against a positive pressure, which may be varied to generate specific exhalation flow rates. Cameli et al. (6) investigated the potential role of FeNO and CalvNO in the diagnosis of CLAD. FeNO was performed in LTx patients, including those with BOS and healthy controls. The study showed higher values of FeNO and CalvNO in LTx patients than in healthy controls. BOS patients showed higher FeNO and CalvNO than non-BOS patients. This suggests that FeNO and CalvNO could be useful markers in the diagnosis of BOS; CalvNO showed higher sensitivity and specificity than FeNO in identifying BOS in LTx patients.

### 3.2. Exhaled Surfactant Protein A

Surfactant protein A (SP-A) is a member of the collectin family and a component of surfactant; it is mostly produced by alveolar type II cells. The main physiological function consists of reducing the surface tension of the alveoli [94,95]. Several studies have demonstrated the importance of SP-A in respiratory infections [96]. A significant reduction in SP-A in BAL fluid has been demonstrated in lung transplant recipients with BOS compared with stable patients [97]. A recent study [59] compared levels of SP-A in exhaled breath of BOS patients, stable LTx patients and healthy controls. The results indicated that SP-A in exhaled particles and the SP-A/albumin ratio were lower in the BOS group than in the BOS-free group. Low levels of SP-A in exhaled particles were associated with an increased risk of BOS.

## 4. Circulating Biomarkers

### 4.1. Serum/Plasma Biomarkers

#### 4.1.1. Lipocalin 2

Lipocalin 2 (LCN2) is a 25 kDa member of the lipocalin protein family, a family sharing a common molecular structure that binds hydrophobic molecules [98,99]. From a pathogenetic point of view, LCN2 has been implicated in modulating apoptosis, showing both pro- and anti-apoptotic activities [100]. In lung transplants, higher serum concentrations of LCN2 have been reported in patients with RAS than in those with BOS. During chronic inflammation of the lung, LCN2 reprograms the immune microenvironment and predisposes tissues to cancer development and progression [101]. Increased serum concentrations of LCN2 and activin-A have also been considered predictors of a lower likelihood of freedom from CLAD in stable LTx patients [42].

#### 4.1.2. Galectins 1 and 3

Galectins are a family of 17 β-galactoside-binding proteins that modulate intracellular signalling through direct interactions with cell adhesion molecules. Galectins are also involved in the regulation of innate and adaptive immune responses [102]. Gal-1 typically functions as an anti-inflammatory and pro-resolving mediator by modulating innate and adaptive immune responses. Serum concentrations of Gal-1 may be dysregulated in various inflammatory scenarios, such as microbial infection, autoimmunity and cancer. Galectin-3 (Gal-3) has been implicated in the development of pulmonary fibrosis. Elevated concentrations of Gal-3 are associated with restrictive interstitial abnormalities, including decreased lung volumes and altered gas exchange [103]. In LTx, a study showed that concentrations of galectin-1 were higher in BOS than in stable LTx patients [45]. According to another study, concentrations of galectin-3 were significantly higher in LTX recipients with airway obstruction than in recipients without any complications [104]. 

#### 4.1.3. Soluble CD59

CD59 is a small glycosylphosphatidylinositol-anchored protein and the sole membrane regulator of the membrane attack complex (MAC). CD59 has been investigated in different diseases as a predictive biomarker of outcome and disease progression [105]. In sepsis, serum concentrations of soluble CD59 (sCD59) were correlated with the severity of organ damage [106]. Budding et al. were the first to investigate the role of CD59 in lung transplant, showing that in chronic lung allograft dysfunction, BOS patients had higher serum concentrations of sCD59 than non-BOS patients [56].

#### 4.1.4. MMP-3

Matrix metalloproteinase-3 (MMP-3) or stromelysin-3 is a protein expressed by different subsets of cells [107]. MMP-3 has been implicated in a range of pathological processes, including acute lung injury, pulmonary fibrosis and lung cancer. It may also promote the breakdown of alveolar epithelial barriers and acute inflammatory responses, particularly in a setting of ventilator-induced lung injury. Genetic deletion of MMP-3 in mice confers protection against bleomycin-induced fibrosis, while transient overexpression of MMP-3 results in profibrotic responses in rat lungs. This presumably occurs by induction of the Wnt-β-catenin pathway by MMP-3. The protein also mediated the degradation of the ECM, enhancing a profibrotic environment, which may affect the phenotype of fibroblasts and promote further deposition of ECM and fibrosis [107,108].

MMP-3 was consistently identified in patients with BOS, suggesting that this protein may be BOS-specific and linked to the development of the disease. In particular, plasma samples from hematopoietic cell transplant recipients with incipient BOS were compared with those of patients with lung infections, chronic graft-versus-host disease without pulmonary involvement and chronic complications after hematopoietic cell transplant. Plasma concentrations of MMP-3 were not elevated prior to the onset of BOS. MMP-3 was only elevated when FEV1 decreased and could, therefore, be a non-invasive tool for diagnosis rather than a prognostic marker [57,109].

#### 4.1.5. MMP-9

Matrix metalloproteinase-9 (MMP-9), also known as 92 kDa type IV collagenase, 92 kDa gelatinase or gelatinase B (GELB), belongs to the zinc-metalloproteinase family involved in the degradation of the extracellular matrix [110]. MMPs play key roles in developing and maintaining adequate oxygenation in health and disease. Broadly, except for MMP2 and MMP14, most deletions in MMPs fail to affect lung development; however, their individual absence can alter the pathophysiology of respiratory diseases. Specifically, under stress conditions, such as acute respiratory infection and allergic inflammation, MMP-9 and others can play a protective role through bacterial clearance and production of chemotactic gradients [111].

In plasma, MMP-9 was associated with BOS and predicted the occurrence of CLAD 12 months before it was diagnosed on the basis of lung function [52,112].

In paper [58] included in our review, levels of MMP-9 were investigated via zymography and gelatin degradation. As expected, BAL concentrations of MMP-9 were elevated in BOS patients together with neutrophil percentage. As demonstrated in other studies [52], MMP-9 contributes to the remodelling processes, leading to airway obstruction. In conclusion, MMP-9 can be considered a prognostic and diagnostic biomarker of BOS. 

There are many studies that also sustain that the epithelial-mesenchymal transition is the main pathogenetic mechanism in BOS. For example, high levels of MMP-9 have been found in BAL of CLAD patients [112], and TGF-beta (a major inducer of epithelial-mesenchymal transition) has been reported at high concentrations in BAL of BOS patients [113].

#### 4.1.6. Self-Antigens (SAgs): K-Alpha 1 Tubulin and Collagen-V

Antibodies to k-α1 tubulin (K-α1T) and collagen type V (Col V) are both associated with the development of CLAD [114]. K-α1T is a gap junction protein with mostly intracellular functions Col V that is usually hidden in the structure of collagen type I in lung tissue extracellular matrix, but when exposed, it can act as an immunogenic ECM protein [115]. Both neo-antigens can be exposed after graft injury, leading to the induction of an immune response that may be aggravated by the loss of peripheral tolerance through immunosuppression of regulatory T-cells [114]. When bound to alveolar epithelial cells, K-α1T antibodies directly influence the onset of airway obliteration by inducing an increase in fibrogenic growth factor expression and fibroproliferation.

Higher serum concentrations of anti-K-α1T and anti-col V after LTx have been associated with a higher risk of BOS in LTx patients. Saini et al. showed that anti-K-α1T levels in serum and BAL fluid were significantly higher in patients with BOS than in those without BOS, matched for time since transplant. BAL concentrations of anti-Col V were also significantly higher in BOS than in non-BOS patients [116]. The occurrence of antibodies directed against self-antigens after LTx was shown to be linked to the development of donor-specific HLA antibodies in the recipient and could be an interaction between allo- and auto-immunity.

Antibodies to Col V and K-α1T were found in 70% of LTx patients who had antibodies against self-antigens after the transplant. Patients with pre-transplant self-antibodies had shorter BOS-free survival than LTx patients who did not have pre-transplant self-antigen antibodies, suggesting that the measurement of these antibodies may aid risk prediction for BOS after LTx [117]. It was recently demonstrated that concentrations of K-α1T were higher in exosomes from LTx patients with BOS [40,118]. Earlier studies also demonstrated that exosomes isolated from LTx patients with BOS contained lung SAgs (Col-V and K-α1T). Exosomes from LTx contained significantly higher levels of Col-V 6 and 12 months before BOS.

Regarding concentrations of HLA-DR and HLA-DQ, Bansal et al. demonstrated that they were significantly higher in exosomes isolated from LTx patients with RAS than in those of patients with BOS. RAS exosomes also contained higher levels of NfkB, 20S proteasome, PIGR, CIITA and Col-V than BOS exosomes. The concentration of K-α1T was higher in exosomes from LTx patients with BOS [40,118]. 

### 4.2. Bronchoalveolar Lavage Fluid Biomarkers

Fibro-bronchoscopy and BAL are procedures currently used in monitoring protocols for LTx recipients. They have provided much data in the search for biomarkers of CLAD in BAL fluid.

#### 4.2.1. Epithelial Cell Death Markers in BAL Fluid

Epithelial cells play an important role in maintaining airflow and in lung defences, both as a physical barrier and through innate and adaptive immune responses [119]. Epithelial injury has been proposed as a mechanism in the pathogenesis of CLAD [120]. Epithelial cell death can be detected by the release of specific intracellular proteins, such as cytokeratins, which are cytoskeletal proteins that maintain the internal organisation and dynamic processes of cells [121]. 

During necrosis and apoptosis, lung epithelial cells release full-length cytokeratin-18 (CK18). The presence of this protein in BAL fluid may indicate processes of apoptosis and necrosis; M30, obtained from the cleavage of CK18, is an expression of cell apoptosis, while M65, which reflects intact CK18 and caspase-cleaved CK18, is related to cell necrosis. 

Levy et al. established possible roles of M30 and M65 in BOS [122]. They collected BAL fluid from these patients routinely for 24 months after transplant. Acute inflammation was indicated in this cohort by >3% neutrophils in the cell count and traces of M30 and M65. The results showed that M30 and M65 were not related to neutrophil count, and that M65 levels were lower in patients with BOS than in those with RAS, while high levels of M65 were associated with worse survival in CLAD patients [67]. These results suggest that M65 could be a useful diagnostic marker of BOS.

#### 4.2.2. Humoral and Cell Immunity Biomarkers in BAL Fluid

Chronic rejection after transplant is predominantly mediated by T cells; however, there is recent evidence that B-cell activation with the production of donor-specific antibodies (DSA) directed against specific human leukocyte antigens (HLA) of the graft is involved [123]. Circulating DSA promotes complement activation, resulting in lung injury [124]. According to the ISHLT definition of antibody-mediated rejection, the presence of these antibodies in BAL fluid is correlated with an increased risk of CLAD [125], especially RAS [126].

In a retrospective study, Vandermeulen et al. [58] investigated DSA in blood in relation to complement activation and immunoglobulins in BAL, as well as the correlation of these humoral markers with airway remodelling. 

The association between K-α1T and SAgs that we previously described in the serum of BOS patients was also analysed in BAL fluid in relation to club cell secreted protein (CCSP). CCSP is an anti-inflammatory protein used as a biomarker for respiratory stress in experimental models of acute and chronic lung injury. Lung transplant patients diagnosed with BOS showed a significant decline in BAL fluid concentrations of CCSP 7–9 months before BOS could be diagnosed clinically. Those who developed SAgs at the time of diagnosis of BOS had lower BAL fluid levels of CCSP than did stable LTx patients without SAgs.

Regarding humoral immunity, higher levels of immunoglobulins, without specific distinctions of sub-classes (IgA, IgM, IgG1,2,3,4) and complements (C1q and C4d), were found in the BAL fluid of patients with AMR overlapping with CLAD and a prevalence of RAS phenotype than in patients with BOS and controls, confirming the finding of Roux et al. [126].

The roles of CD4+ and CD8+ cells in solid transplant patients are still unclear. Activation of these cells in acute cell rejection after lung transplant [127] made it necessary to understand their importance in chronic rejection [97]. Hayes et al. [60] described the BAL fluid T cell panel in LTx patients affected with BOS. They demonstrated that LTx patients without BOS did not show changes in CD4+ and CD8+ cells in BAL samples within two years of transplant. On the contrary, recipients who developed BOS showed a decline in CD4+ and an increase in CD8+ cells, with a significant decline in the CD4:CD8 ratio in the same matrix in the same period. Subject to the small population of this study, the T cell profile could be considered a possible biomarker for BOS and probably for other CLAD phenotypes. A recent paper also demonstrated a predominance of Th17 cell subtypes and a depletion of Tregs and Bregs in the BAL fluid of patients with acute rejection with respect to BOS patients [43]. 

#### 4.2.3. Cytokines and Chemokines in BAL Fluid

In lung transplant patients, cytokine production occurs in two steps: first, in an early antigen-independent cascade triggered by the recipient’s immune system, surgical trauma, donor lung status or ischemic reperfusion injury; second, in a late antigen-dependent cascade that directs activated recipient lymphocytes into the graft [128,129,130]. The balance between the production of pro- and anti-inflammatory cytokines and chemokines is critical for airway repair and, therefore, also for the development of CLAD [131]. Berastegui et al. [54] examined the production of cytokines in lung transplant recipients who developed CLAD. They found that IL-5 seemed to be the most significant biomarker, being more highly expressed in RAS than in BOS. A study by Itabashi et al. also reported that the loss of CCSP may contribute to increased production of IL-8, suggesting that CCSP plays an important role in regulating the proinflammatory cascade [44].

#### 4.2.4. Gene Expression in BAL Fluid

The common rejection module is composed of 11 genes (*CD6*, *TAP1*, *CXCL10*, *CXCL9*, *INPP5D*, *ISG20*, *LCK*, *NKG7*, *PSMB9*, *RUNX3* and *BASP1*) that are overexpressed during allograft dysfunction. It has been studied in many types of solid transplants to quantify inflammation in tissues [132]. Sancreas et al. [66] developed a study to understand how the module could have a biomarker role in acute rejection and CLAD. They performed BAL, trans-bronchial biopsy and histological studies in explant organs, discovering that the expression of these genes is higher in acute rejection than in chronic rejection. The most relevant circulating biomarkers and gene expression are represented in Figure 2.

#### 4.2.5. NMR Spectroscopic Detection of Metabolites in BAL Fluid

There have been few studies on the possible utility of NMR spectroscopy in respiratory diseases. Some have focused on the mouse respiratory system [133], on preterm infants with respiratory distress syndrome [134] and on paediatric patients with cystic fibrosis, in search of a correlation between the degree of airway inflammation and the number of metabolites in BAL fluid [135]. Although little information is available about the metabolic profile of BAL fluid in adults, and even less in BOS patients, Ciaramelli et al. developed a pilot study to assess the suitability of using NMR spectroscopy to explore the metabolic profile of the BAL fluid of lung transplant recipients with or without BOS [55]. If this method could predict BOS, early intervention to prevent irreversible lung damage may be possible.

NMR is able to detect different metabolites, such as amino acids, Krebs cycle intermediates, mono- and disaccharides, nucleotides and phospholipid precursors. The authors tried to understand whether some of these molecules could be early biomarkers of BOS. They found high levels of some branched-chain amino acids (valine, leucine, isoleucine) in the BAL fluid of BOS patients, and these levels were related to the severity of BOS. 

Another potential role is played by the taurine/hypotaurine pathway. Taurine is a regulator of cell volume via membrane stabilisation and has antioxidative, anti-inflammatory and anti-apoptotic properties [136]. Taurine is a weak agonist of chloride-permeable gamma-aminobutyric acid type A receptors (GABA-A R) and glycine receptors (GlyR) that are located not only in neural synapses but also in the central nervous system and the lungs. This suggests that taurine plays a role in the lungs, potentiating the relaxation of airway smooth muscle cells through binding to GABA-A R and secretion of mucus by goblet cells [137]. The release of ROS by neutrophils and macrophages during inflammatory states could stimulate a decrease in the uptake of intracellular taurine and an increase in the taurine release into extracellular fluids, with a consequent increase in mucus secretion. Increased levels of taurine in the BAL fluid of BOS patients may, therefore, reflect inflammation-induced osmotic stress or epithelial cell damage. Since taurine accumulates in the cytoplasm of neutrophils and other leukocytes, its presence may also suggest high levels of leukocytes in the airways (due to a state of inflammation) [138]. Moreover, levels of lactate in BAL fluid are related to an inflammatory state. BOS patients with increased BAL fluid levels of lactate could have an inflammatory status that exacerbates the disease. Together these results suggest that there are possible markers for early diagnosis of BOS.

## 5. Extracellular Vesicles 

### 5.1. LKB1 from Tissue

Downregulation of *LKB1* has been demonstrated in circulating exosomes prior to clinical diagnosis of BOS, suggesting that this enzyme may have a role in the pathogenesis of the syndrome. In particular, incubation of BOS-exosomes also decreased *LKB1* expression and induced epithelial-mesenchymal transition markers in an air–liquid interface culture method [38]. This study provided new evidence that exosomes released from transplanted lungs undergoing chronic rejection are associated with an inactivated tumour suppressor gene, *LKB1*, and this loss induces epithelial-mesenchymal transition, leading to CLAD in humans.

### 5.2. miRNAs

Recent studies suggest that epigenetic regulation of microRNAs might play a role in the development of BOS. Di Carlo S. et al. found, through in situ hybridisation, the dysregulation of two candidates, miR-34a and miR-21, as pathogenetic factors of BOS [139]. Xu Z. et al. identified miR-144 as the most significant altered miRNA. miR-144 is a principal critical regulator of TGF-β signalling, involving an increase in Smad-2, Smad-4, FGF and VEGF, leading to diminished fibrogenesis [140]. 

Recently, the alteration of Smad-4 has been found in association with miR-155-5p and miR-23b-3p, which resulted in an altered expression in responders to extracorporeal photopheresis [141].

Budding et al. showed that miR-21, miR-29a, miR-103 and miR-191 levels were significantly higher in BOS+ patients prior to clinical BOS diagnosis [142]. Most studies have focused on extracellular miRNAs as potential biomarkers since they are stable and can be detected in blood, urine and other body fluids by simple, sensitive and relatively cheap assays, even after years of sample storage. miR-21 and miR-29a are the most commonly observed aberrant miRNAs in human cancers. Post-transcriptionally, miR-21 downregulates the expression of the tumour-suppressor gene *PTEN* and stimulates growth and invasion in non-small cell lung cancer. Inhibition of miR-21 reduces cancer cell proliferation, migration and invasion. miR-103 has recently been associated with the metastatic capacity of primary lung tumours [143]. 

Bozzini S. et al. suggested that miR-21-3p has been identified in the context of OB fibrogenesismiR-21-3p playing a role as a profibrotic effector in CLAD fibro-obliterative processes [144]. 

## 6. Whole Blood and BAL Cell Subsets

To avoid the onset of BOS, it is important to define the cause of inflammation and to understand how immune cells are implicated in inflammation and/or repair. Immunosuppression therapy is the major management strategy for preventing the rejection of the new organ by activation of the immune system [145]. Patients receive a regimen of immunosuppressants that reduces the differentiation and proliferation of lymphocytes, including B and T cells [146].

### 6.1. B Cells

Many recent studies have focused on the role and association between long-term allograft survival and B cells. B lymphocytes are a cell population that expresses clonally different cell-surface immunoglobulin (Ig) receptors that recognize specific antigen epitopes [147]. Phenotypically, B cells are differentiated on the basis of the expression of specific cell-surface markers [148]. Transitional B cells are the most immature peripheral B cells and express CD24 and CD38. Whereas human memory B cells can be divided into different populations depending on the expression of markers IgD and CD27: naïve (IgD+CD27-), non-switched memory (IgD+CD27+), switched memory (IgD-CD27+) and CD27-negative (IgD-CD27-) B cells, also known as double-negative B cells (DN B cells). According to the literature, memory B cells and plasmablasts are correlated with early acute antibody-mediated rejection [149], while naïve and transitional B cells are associated with tolerance in kidney and liver transplants [150].

The peripheral B cell subset has been investigated in lung transplant recipients with and without BOS and in healthy controls. An increase in DN B cells was recorded in patients with BOS, along with a decrease in transitional and naïve B cells in BOS patients with respect to healthy controls [151]. The medication used to prevent the development of allograft dysfunction usually prevents the proliferation of B cells.

During BOS, the immune system is exposed to abundant antigens due to chronic inflammation and repair processes [152]. It is probable that naïve and transitional B cells differentiate, and their peripheral frequency decreases while DN B cell numbers increase.

Regulatory B cells (Bregs) function as immunosuppressors through the production of anti-inflammatory cytokines, such as IL-10, IL-35 and transforming growth factor β (TGF-β) [153]. Regarding B cells, CD19+CD24hiCD38hi are reported to be elevated in the BAL of acute rejection patients [154]. They suppress immunopathology by prohibiting the expansion of pathogenic T cells and other proinflammatory lymphocytes. Regulatory B cells support immunological tolerance and may be involved in maintaining long-term allograft function.

CD9 is a cell surface glycoprotein of the tetraspanin family, characterized by four transmembrane-spanning domains and two extracellular domains [155]. Several studies implicate CD9 in the immunosuppressive activity of Bregs [156].

Brosseau et al. [68] demonstrated the importance of Bregs and the CD9 marker, a powerful predictive marker of stability and long-term BOS-free survival after lung transplant. Twenty-four months after transplant, plasma concentrations of CD9 were higher in stable patients than in those who would develop BOS. CD9 B-cell frequencies allow excellent discrimination between stable patients and those destined to develop BOS at 24 months.

### 6.2. T Cells

CD8+ and CD4+ T cells are two subpopulations of the adaptive immune system: CD4+ T cells are helper T cells that assist other lymphocytes in evoking an immune response [157]. CD8+ T cells are cytotoxic T cells that kill virus-infected cells and tumour cells [158]. They are also involved in transplant rejection. Durand et al. [48] demonstrated that the CD4+ and CD8+ compartments are not biased in BOS patients. An increased proportion of circulating CD4+CD25hiFoxP3+ T cells was reported one month after LTx in patients who proceeded to develop BOS within 3 years. These cells may act as negative regulators of BOS development. Although this result is encouraging, the underlying mechanisms leading to the increased proportion of regulatory CD4+CD25hiFoxP3+ T cells in BOS patients remain unclear.

Regulatory T cells modulate the activation and proliferation of other immune cells and are crucial for maintaining T-cell tolerance to self-antigens. They secrete different immunosuppressive cytokines, such as IL-10 and TGF-β [134,135].

Piloni et al. [50] analysed the long-term peripheral kinetics of Tregs in lung transplant patients and assessed their association with different clinical variables. Previous studies demonstrated that peripheral Tregs may be a major regulatory subset in lung transplant. The latest results confirmed the role of Tregs in lung graft acceptance/rejection. Peripheral Treg counts fell significantly in CLAD patients. The degree of this decrease was correlated with the severity of BOS.

Th17 cells secrete IL-17, IL-22 and IL-23, and recruit neutrophils to sites of infection. In this way, they cause inflammation and autoimmunity [159,160,161].

In order to determine the phenotype of Tregs, Bregs and Th17 cells, samples of peripheral blood from stable lung transplant patients were analysed [43]. An increase in Th17 subtype percentages in PBMCs and BAL fluid and a reduction in Tregs and hence in the Treg/th17 ratio were observed in acute rejection patients. It was demonstrated that cytotoxic and proinflammatory lymphocyte percentages increased after transplant [162]. To explore their role, Hodge et al. [65] measured intracellular cytotoxic mediator granzyme B, interferon-gamma, tumour necrosis factor (TNF), alpha proinflammatory cytokines and CD28 in blood, BAL fluid and large airways in patients with BOS. The results indicated a significant decrease in T cells, an increase in NKT-like cells and an increase in CD8+ T and NKT-like cells in BOS patients with respect to stable patients and healthy controls. In detail, the percentages of senescent CD28null CD8+ T and CD8- T and NKT-like cells increased in BAL fluid and in large and small airways in patients with BOS. Loss of CD28 was associated with more T and NKT-like cells expressing granzyme B, IFN-γ and TNF-α. Loss of CD28 seemed to be associated with repeated antigen stimulation, as demonstrated in chronic obstructive pulmonary disease [163]. A good idea may be to find some treatment options targeting the proinflammatory nature of these cells.

### 6.3. Monocyte-Macrophage Lineage

The monocyte-macrophage lineage plays an important role in the immunopathology of chronic rejection [164]. Monocytes recognize “danger signals” via pattern recognition receptors. They conduct phagocytosis, present antigens, secrete chemokines and proliferate in response to infection and injury.

Human blood monocytes can be divided into three subsets, classical, intermediate and non-classical, based on CD16 and CD14 expression [165]. To distinguish the non-classical and intermediate subsets, another marker, 6-sulfo LacNAc (SLAN), has been identified. Described for the first time as a cell-surface marker of a certain type of dendritic cell in human blood, SLAN is a carbohydrate residue associated with the non-classical monocyte subset [166]. The functions of monocyte subsets are different and also depend on the inflammatory pathology. In the lungs, the classical subset can differentiate into pulmonary dendritic cells and macrophages, whereas the functions of the other two subsets are still unknown [167].

Schreurs et al. analysed peripheral monocyte subsets in LTx patients with and without BOS [61] and found a reduction in the number of non-classical monocytes, i.e., those expressing SLAN-positivity, in the former, whereas the expression of SLAN-positivity was higher in BOS patients. To highlight the activity of T cell co-stimulation and antigen presentation, they also stained blood monocytes for surface markers HLA-DR and CD86. The only monocyte marker that proved significant in BOS was HLA-DR, which showed a significant increase in expression on non-classical monocytes. Although BOS is an outcome of chronic inflammation, it does not lead to increased production of monocytes. Further studies will be necessary to validate these results.

### 6.4. Whole Blood Gene Expression Profiles

Performing a non-invasive microarray gene expression profiling of whole blood, Danger et al. [49] identified and validated *POU2AF1* (POU Class 2 Homeobox Associating Factor 1), *TCL1A* (T-cell leukaemia/lymphoma protein 1A) and *BLK* (B-lymphoid tyrosine kinase) as three predictive biomarkers of BOS more than 6 months before diagnosis. By monitoring these three genes, it proved possible to stratify patients on the basis of BOS risk.

*POU2AF1* is a B cell transcriptional coactivator implicated in B cell development and function [168]. It is also expressed in T cells and functions as a T-cell-dependent B cell response. *BLK* encodes a non-receptor protein, tyrosine kinase, and is involved in the regulation of B cell receptor signalling [169]. *TCL1A,* known as an oncogene, acts as a coactivator of serine-threonine kinase Akt and promotes cell survival, growth and proliferation [170]. *TCL1A* has been reported to be downregulated in the peripheral blood of patients with BOS before disease onset. This is in line with the findings of another study [171], where the gene was downregulated at the time of acute allograft rejection of transplanted kidneys but overexpressed in tolerant patients. The exact contribution of these genes to the development of BOS remains to be investigated.

## 7. Cell Culture Supernatants

### Matrix Metalloproteinase 9

The pathogenesis of BOS remains largely unknown. We can observe the histological characteristics of the lung tissue of LTX patients. BOS is characterized by the infiltration of lymphocytes into the bronchiole walls, followed by a fibrotic process. This leads to the deposition of the extracellular matrix (ECM) that obliterates the small airways [172]. It has been postulated that this fibrotic process begins with epithelial damage caused by alloimmune and non-alloimmune mechanisms, followed by unsuccessful repair. Fibroblasts play a central role in this process, although it is unclear whether they are activated in situ or recruited from circulation. Another role in fibrosis is played by bronchial epithelial cells (BECs) and the epithelial-mesenchymal transition, which leads to the downregulation of epithelial markers (e.g., E-cadherin) and upregulation of mesenchymal markers (e.g., vimentin) [173]. Bronchial epithelial cells, in the process of mesenchymal differentiation, produce matrix metalloproteinases and ECM proteins (including type-I collagen and fibronectin). Matrix metalloproteinase (MMP)-9 plays an important role in all diseases based on air-remodelling processes. According to this pathogenetic explanation of BOS, the allogeneic immune response is the start of the process, but the effect of T cells on BECs is unclear. To demonstrate the role of allogenic immunity via epithelial-mesenchymal transition in BOS, Pain et al. analysed the response of BECs stimulated by T-cells in synergy with TGF-beta [52]. They set out to detect MMP-9 in the cell culture supernatant of LTx patients. Concentrations of MMP-9 were high 12 months before lung function diagnosis, suggesting the role of this metalloproteinase as an early biomarker of CLAD. However, MMP-9 did not distinguish RAS and BOS.

## 8. Radiological Markers

### 8.1. HRCT Quantitative Image Analysis Score

Quantitative image analysis (QIA) of the lung parenchyma could be a useful tool for phenotyping CLAD patients. It could also have prognostic value in terms of survival [39]. The QIA score is based on the proportion of lung volume affected by interstitial disease or air-trapping. It is an imaging scoring system that can be performed on lung CT images by estimating the percentage of Quantitative Ground Glass (QGG), Quantitative Lung Fibrosis (QLF) and Quantitative Honey Combing (QHC). QHC+QLF+QGG is a measure of total interstitial disease (QILD). Air-trapping is then scored by analysing CT scans during end-expiration or residual volume.

Weigt et al. used the QIA score in LTx patients to verify its possible utility for phenotyping CLAD [39]. They analysed chest HRCT images of bilateral LTx patients within 90 days of CLAD onset and in a control (no-CLAD) group. They found that BOS cases had a lower score for interstitial disease than patients with RAS or mixed CLAD phenotypes. However, BOS showed more air-trapping. These results were also compared with the standard-phenotyping system (2019 ISHLT consensus). There was quite a good concordance between the two methods. Another important result was that the risk of death was higher in RAS and mixed phenotypes (with higher QILD) than in BOS. The QIA score on HRCT images can, therefore, be considered a diagnostic as well as a prognostic marker.

### 8.2. Pleural Thickening Evaluated by Lung Ultrasound

Pleural thickening may be focal or diffuse, and it may have different causes, both benign and malignant [174]. In interstitial lung disease, the lung parenchyma and pleura show increased tissue density, which can be demonstrated by lung ultrasonography as a significant number of B-line reverberation artefacts indicative of interstitial syndrome and pleural thickening. RAS is characterized by typical radiological alterations of interstitial lung disease that are absent in BOS, namely pleural thickening and an upper lobe-dominant fibrotic pattern [155].

By virtue of its sensitivity and specificity, lung ultrasonography can be a supplementary tool for identifying pleural thickening on upper lobes useful for phenotyping CLAD patients.

In 2015, Davidsen et al. demonstrated the utility of lung ultrasonography for identifying pleural thickening in CLAD patients, assuming pleural thickening to be a marker of RAS, absent in BOS [41]. They studied BOS and RAS prospectively by chest HRCT and lung ultrasonography. Pleural thickening was more pronounced in RAS, as were chest HRCT features, demonstrating the potential utility of ultrasonographic features and radiological markers to discriminate BOS from RAS.

### 8.3. Parametric Response Mapping of Functional Small Airway Disease by CT

Parametric response mapping (PRM) is a CT method that can be used to quantify functional small airway disease and parenchymal disease. In BOS, the predominant pathogenetic mechanism is associated with small airway disease and, consequently, with airflow limitation [175,176]. Belloli et al. compared the concordance between spirometry data of bilateral lung transplant recipients and PRM results [53]. Patients with isolated FEV1 decline had significantly worse functional small airway disease than their control subjects, whereas patients with declines in FEV1 and FVC (less than 80% at baseline) had worse parenchymal disease. PRM could be useful for phenotyping CLAD patients. Moreover, functional small airway disease >30% proved to be a strong predictor of survival. Patients with these radiological characteristics lived, on average, 2.6 years less than patients with functional small airway disease <30%, suggesting a prognostic role for this marker. Figure 3 represents the radiological and tissue markers of BOS.

## 9. Future Directions

Several markers have already been investigated in CLAD in different specimens, including serum, plasma, cells and tissue. Recently, clinical, functional and radiological scores have been made for the prediction of BOS. Considering that a single biomarker with appropriate accuracy and reliability able to predict the diagnosis and prognosis of BOS is not available, the purpose will be implementing a panel of biomarkers from the biological origin with radiological and/or clinic-functional findings. Furthermore, other biomarkers that showed a good correlation with the pathogenesis of BOS (such as miRNA and gene expression profile) are largely unexplored. Moreover, the idea of using biomarkers of other fibrotic lung diseases (such as IPF) may help to find a novel set of proteins with diagnostic and prognostic features.

## Figures and Tables

**Figure 1 biomedicines-10-03277-f001:**
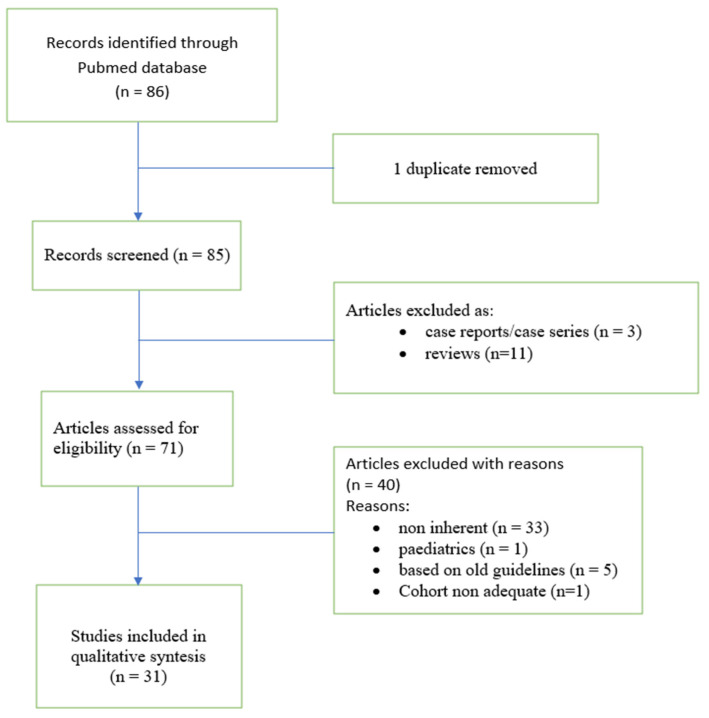
Flowchart of the study.

**Figure 2 biomedicines-10-03277-f002:**
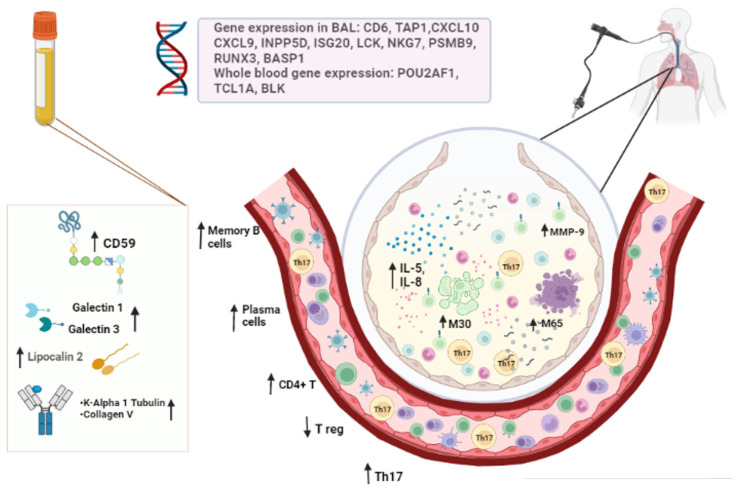
The most relevant circulating biomarkers and gene expression.

**Figure 3 biomedicines-10-03277-f003:**
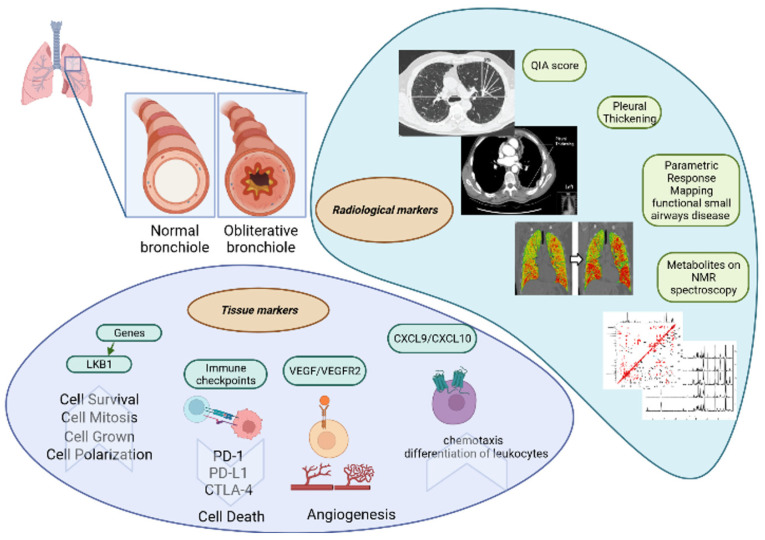
The radiological and tissue markers of BOS.

**Table 1 biomedicines-10-03277-t001:** Characteristics of the selected papers.

Author (Year)	Study Type	Sample Size	Principal Results	Markers	Matrices
**Rahman M, et al.** **Am J Transplant. 2021** [38]	Retrospective	20 BOS and 20 stable	Exosomes released from lung tissue of BOS are associated with inactivated LKB1 that induces EMT	LKB1 gene	Tissue
**Weigt SS, et al. Transplantation. 2021** [39]	Prospective	15 BOS, 13 RAS, 4 mixed CLAD, 15 undefined CLAD phenotypes.	BOS had more air-trapping, lower quantitative image analysis (QIA) score than RAS	QIA score at HRCT	Imaging
**Bansal S, et al. J Heart Lung Transplant 2022** [40]	Retrospective	18 BOS, 13 RAS, 5 stable	Increased levels of circulating exosomes containing HLA-DR and HLA-DQ, NFkB, CIITA, 20S proteasome and PIGR can differentiate between the RAS and BOS	circulating exosomes	Plasma
**Davidsen JR et al. J Clin Med 2021** [41]	Prospective	19 BOS, 6 RAS	Pleural thickening more pronounced in RAS than BOS	Pleural thickening	Imaging
**Veraar C, et al.** **Sci Rep. 2021** [42]	Prospective	11 RAS, 30 BOS56 StableLT	RAS had higher Lipocalin-2 compared to BOS and Stable-LTx	Lipocalin-2	Serum
**Bergantini L, et al. Cells 2021** [43]	Retrospective	6 BOS, 6 acute rejection, 4 stable	Decreased Treg cell BOS than acute and stable. Increase in Th1 and a decrease in Th17 in BAL of BOS patients than RAS and stable	Regulatory and effector cells	Peripheral blood and BAL
**Itabashi Y, et al. Transplantation 2021** [44]	Retrospective	18 BOS, 34 stable	BOSs have lower CCSP levels up to 9 months before BOS diagnosis	CCSP	BAL fluid
**d’Alessandro M et al.** **Lung 2021** [45]	Retrospective	10 BOS, 9 stable, 9 HC	Galectin-1 was higher in BOS than in stable LTx patients	Galectin1-3	Serum
**Righi I et al. Front.Immunol 2021** [46]	Retrospective	4 RAS, 4 BOS, 6 HC	Higher T lymphocytes expressing PD-1, PD-L1 and CTLA4 immune checkpoint.	PD-1, CTLA-4, CD4, CD8, TIGIT.	Tissue
**Cameli P et al. Respir Med 2015** [47]	Retrospective	30 HCs,15 stable, 12 BOS	BOS patients had higher FeNO at 150 and 350 mL/s and CalvNO than non-BOS patients.	FeNO, CalvNO eCO	exhaled
**Durand M et al.** **J Heart Lung Transplant 2018** [48]	Prospective	50 stable, 32 BOS	An increase in Treg T-cells in BOS patients at 1 and 6 months after transplantation.	Treg-cells	Peripgeral blood
**Danger R, et al.** **Front Immunol, 2018** [49]	Clinical trial	62 stable, 43 prediction group, 32 BOS	POU, *POU2AF1*, *TCL1A* and *BCLK* were validated as predictive biomarkers of BOS more than 6 months before diagnosis	POU, *POU2AF1*, *TCL1A* and *BCLK* gene	Peripheral blood
**Piloni D et al.** **BMC Pulm Med 2017** [50]	Retrospective	137 LTx	T-reg cell counts progressively decreased according to the severity of CLAD	T reg cells.	Peripheral blood
**Budding K, et al. TRanspl. Immun. 2017** [51]	Clinical trial	10 BOS, 10 without BOS	MiR-21, miR-29a, miR-103 and miR-191 levels were significantly higher in BOS patients prior to clinical BOS diagnosis	miRnas	Serum
**Pain M, et al.** **Am J Transplant 2017** [52]	Clinical trial	49 stable, 29 BOS, 16 RAS	Plasma MMP-9 was associated with BOS and predicted the occurrence of CLAD, 12 months before the functional diagnosis	Matrix metalloproteins-9	Culture supernatants and plasma
**Belloli EA, et al.** **Am J Respir Crit Care Med 2017** [53]	Clinical trial	52 LTx with spitometric decline	FEV_1_ decline has significantly higher PRM^fSAD^ than their control	PRM^fSAD^	Imaging
**Berastegui C. et al.** **Clin Transplant 2017** [54]	Cross-selectional study	15 BOS, 7 RAS, 29 stable	BALF neutrophilia were higher in BOS than in stable. IL-5 presented significant differences between BOS and RAS	IL-5	BAL Fluid
**Ciaramelli C. et al.** **J. Proteom Res, 2017** [55]	Retrospective study and pilot study	10 stable, 10 BOS 0p, 10 BOS	Suitability of the NMR approach in monitoring different pathological lung conditions	Metabolites	Imaging
**Budding K, et al.** **Sci Rep. 2016** [56]	Case-control	20 BOS, 69 non-BOS, 20 HC	BOS patients showed higher sCD59 diagnosis compared to non-BOS patients	sCD59	Serum
**Liu X, et al.** **Am J Transplant. 2017** [57]	Retrospective	12 BOS, 16 infection, 15 CGVD without pulmonary involvement	Fibronectin 1 (FN1) and MMP-3 were associated with BOS development	FN1 and MMP-3	Plasma
**Vandermeulen E, et al.** **Transpl. Immunol. 2016** [58]	Retrospective	15 BOS, 16 RAS, 14 stable	Increased levels of immunoglobulins and complement proteins are dominantly present in CLAD	Immunoglobulins C4d and C1q, MMPs	BAL Fluid, Tissue Peripheral blood
**Ericson PA, et al.** **Transplant Direct. 2016** [59]	Case-control	26 stable, 7 BOS, 33 HC	SP-A in exhaled particles and the SP-A/albumin ratio were lower in the BOS group compared to the BOS-free group	Surfactant protein A and the SP-A/albumin ratio	Exhaled
**Hayes Jr D, et al.** **Lung. 2020** [60]	Case-control	6 non-BOS, 10 BOS	A decline in CD4+ T-cell and increase in CD8+ T-cell in BOS when comparing baseline values and at 6 months follow-up	T-cell subsets	BAL Fluid
**Schreurs I, et al.** **Transplant Proc. 2020** [61]	Case-control	27 LTx, 17 BOS, 10 HC, 9 sarcoidosis patients	The absolute count of monocytes was decreased in BOS. The expression of both CD36 and CD163 was significantly increased in the LTx and BOS cohort	CD36 and CD163	Peripheral blood
**Schreurs I, et al.** **Exp Clin Transplant. 2020** [62]	Clinial trial	17 BOS, 10 HC	Transitional and naïve B cells were decreased in BOS.	B-Cell subsets	Peripheral blood
**Larsson-Callerfelt AK, et al.** **Respir Med 2020** [63]	Case-control	14 LTx, 11 BOS	VEGF were lower 3 months after lung transplantation compared to non-transplanted subjects	VEGF, VEGF 2 receptor	Tissue
**Sharma M, et al.** **J Heart Lung Transplant 2020** [64]	Retrospective	21 with BOS, 10 stable	Circulating exosomes isolated from BOS demonstrated increased levels of lung SAgs (Kα1T and Col-V) 12 months prior to the diagnosis	Kα1T and Col-V	Plasma
**Hodge G, et al.** **J Heart Lung Transplant. 2021** [65]	Case-control study	10 BOS, 11 stable and 10 HC	Increased CD28null T and NKT-like cells were identified in BOS compared with that in the controls and stable transplant recipients	Granzyme B, IFN-γ, TNF-α, CD28	Peripheral blood, Bal Fluid
**Sacreas A, et al.** **PLoS One. 2018** [66]	Retrospective	13 stable, 8 AR, 9 BOS, 10 RAS.	Transcriptional tissue analysis for CLAD distinguishes RAS from BOS.	*CD6*, *TAP1*, *CXCL10*, *CXCL9*,	BAL, Tissue
**Levy L, et al. Transpl Int. 2019** [67]	Retrospective	10 RAS, 16 BOS and 19 long-term CLAD-free controls.	M65 levels were significantly lower in BOS compared to RAS. Detection of BAL M65 may be used to differentiate CLAD subtypes	M65, fragments of cytokeratin-18	BAL
**Brosseau C, et al.** **Am J Transplant. 2019** [68]	Retrospective	18 stable, 16 BOS	CD24hi CD38hi transitional B cells were increased in stable patients. These CD24hi CD38hi transitional B cells displayed significantly higher incidence of BOS	B cell profile	Peripheral blood

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
