# Peer review of "Markers of Bronchiolitis Obliterans Syndrome after Lung Transplant: Between Old Knowledge and Future Perspective"

_biomedicines, 2022, doi:10.3390/biomedicines10123277_

Round 1

Reviewer 1 Report

GENERAL

The authors have performed a careful review of the available literature discussing markers of CLAD and specifically BOS. In the main the review is comprehensive, well balanced and adequately reports the state of the art. An executive summary including future directions would be useful as would additional insights gained from the pathogenesis of obliterative bronchiolitis.

MAJOR

1. Despite mentioning the ISHLT Consensus Definition  of CLAD and RAS it is puzzling that the actual ISHLT manuscripts are not cited, rather secondary texts, which perhaps explains a lack of precision in the definition of CLAD and BOS in particular. Indeed a careful read of these two manuscripts should lead to further insights which would correct some of the somewhat broad and loose generalisations about BOS that populate the Review  (eg see next two points). Capitals should be used for the spelled out ISHLT.

2. RAS has a worse prognosis than BOS and the statement regarding BOS should therefore be adjusted.

3. BOS is a syndrome defined by a change in lung function, most commonly due to obliterative bronchiolitis. Saying that BOS is a syndrome that leads to pathological changes fails to understand what constitutes a syndrome. An in depth reappraisal of the literature regarding obliterative bronchiolitis would prove informative to improve the quality of the review.

4. Line 76 describes the "molecular" point of view but then describes the histopathological changes...

MINOR

1/. Reference #3 is cited again as Reference #151

2/. line 187, breath is not "reproducible" although its analysis may be..

Author Response

Thank you the reviewer for the comments. We attached below the response to the comments.

Reviewer 2 Report

The manuscript is interesting and well written. I believe that it takes into consideration most of the data present in the literature on the subject.

I would ask the authors to expand the part on miRNAs, also in consideration of the fact that some papers on circulating and tissue levels of miRNAs in BOS have recently been published. I would also recommend adding even some small relationship to possibly implicated pathways, in addition to PTEN.

Author Response

Thank you the reviewer for the comments.

I attached below the response to comments.
